# Diagnosing the performance of food systems to increase accountability toward healthy diets and environmental sustainability

Anna Herforth[1], Alexandra L. Bellows[2], Quinn Marshall[2], Rebecca McLaren[3], Ty Beal[4], Stella Nordhagen[5], Roseline Remans[6], Natalia Estrada Carmona[6], Jessica Fanzo[2,3,7] *

1 Harvard T.H. Chan School of Public Health, Boston, Massachusetts, United States of America, 2 Bloomberg School of Public Health, Johns Hopkins University, Baltimore, Maryland, United States of America, 3 Berman Institute of Bioethics, Johns Hopkins University, Baltimore, Maryland, United States of America, 4 Global Alliance for Improved Nutrition, Washington, DC, United States of America, 5 Global Alliance for Improved Nutrition, Geneva, Switzerland, 6 The Alliance of Bioversity International and CIAT, Montpellier, France, 7 Nitze School of Advanced International Studies, Johns Hopkins University, Washington, DC, United States of America

* jfanzo1@jhu.edu

**Data Availability Statement:** All data are available on the Food Systems Dashboard https://foodsystemsdashboard.org/. These are these third party data. The author have no special access to

## Abstract

To reorient food systems to ensure they deliver healthy diets that protect against multiple forms of malnutrition and diet-related disease and safeguard the environment, ecosystems, and natural resources, there is a need for better governance and accountability. However, decision-makers are often in the dark on how to navigate their food systems to achieve these multiple outcomes. Even where there is sufficient data to describe various elements, drivers, and outcomes of food systems, there is a lack of tools to assess how food systems are performing. This paper presents a diagnostic methodology for 39 indicators representing food supply, food environments, nutrition outcomes, and environmental outcomes that offer cutoffs to assess performance of national food systems. For each indicator, thresholds are presented for unlikely, potential, or likely challenge areas. This information can be used to generate actions and decisions on where and how to intervene in food systems to improve human and planetary health. A global assessment and two country case studies—Greece and Tanzania—illustrate how the diagnostics could spur decision options available to countries.

## Introduction

Food systems include the people, places, and methods involved in producing, storing, processing and packaging, transporting, and consuming food; they can consist of either long or short supply chains and be global or local [1, 2]. Food systems have the potential to yield multiple positive outcomes including delivering healthy diets that protect against multiple forms of malnutrition and disease; safeguarding environments, ecosystems, and natural resources; and supporting fair, equitable livelihoods [3–5]. However, food systems are currently managed and governed in ways that do not meet these outcomes as well as they could [6–8].

these data and the data comes from open access, widely used sources such as World Bank and FAO Stat.

**Funding:** This research was supported by a grant from The Rockefeller Foundation, Grant number 2020 FOD 021. https://www.rockefellerfoundation.org/ AH, QM, RM, TB, SN, RR, and JF received funding from this award. There was no funding for the study or authors from commercial companies. The funders had no role in study design, data collection and analysis, decision to publish, or preparation of the manuscript.

**Competing interests:** The authors have declared that no competing interests exist.

More specifically, approximately three billion people cannot afford a healthy diet, and an estimated 738 million are hungry and unable to access sufficient dietary energy [9]. At the same time, the trade and sales of ultra-processed foods (UPFs) high in salt, added sugars, refined flours, and unhealthy fats are increasing, associated with poorer nutrient profiles of diets and adverse health effects [10–14]. Poor diets are associated with malnutrition in all its forms, including as a cause of nutrient deficiencies and undernutrition and as a risk factor for increased deaths and cardiovascular disease, diabetes, and some cancers [15]. Current extractive food systems are unsustainably using land and water resources while contributing 21 to 37% of global greenhouse gas emissions (GHGe) [16, 17]. Agricultural land use occupies five billion hectares, with 1.5 billion hectares used for cropland and 3.5 billion for grazing land [18]. This use accounts for 40% of the ice-free land mass [19]. These large land requirements drive deforestation and loss of biodiversity while producing GHGe including carbon dioxide, nitrous oxide, and methane [6]. In addition, food systems account for 70 to 80% of freshwater consumptive use [20, 21] and are currently a source of significant soil degradation, air and water pollution, and solid waste [22–25].

Food systems can be transformed to reduce negative impacts on human health and the environment with comprehensive policies, increased investments, and enhanced risk management [26–29]. Food systems; however, are complex, and human and environmental health outcomes related to food systems are multi-faceted. As a result, it is difficult to have a clear picture of how and where to act to target specific challenge areas in any given setting [30]. In order to make sound investments, decision-makers need information on the current state of their food systems and how they relate to food security, nutrition, health, and environmental outcomes. In short, there is a need for better diagnostics of food systems to strengthen food systems governance and accountability [31].

The Food Systems Dashboard (FSD) was launched in 2020 to provide a single platform for food systems data relevant to diet and environmental outcomes, and to enable the use of these data for policymaking [32]. The FSD is intended for policymakers, non-governmental organizations, civil society leaders, educators, researchers, businesses, and other actors to enable timely visualization of national food systems and compare across countries, regions, income classifications, and food system types. It combines data for over 200 indicators from over 40 sources, for more than 230 countries and territories (about 630,000 data points). This information is organized into major components of food systems, including agricultural production and supply chains; food environments; diets, nutrition, and health; environmental outcomes; and socio-ecological drivers of food systems.

This paper aims to build the parameters for diagnosing likely challenges within food systems, using the data assembled in the FSD. This diagnosis will aid the interpretation of food systems data, so that decision-makers can see what is going relatively well and what is challenging in each setting and consider a range of possible actions to address challenges and maintain successes. First, out of all 200 indicators currently on the FSD, a set of indicators with diagnostic value is identified. Second, cutoff points to diagnose likely challenges are discussed and proposed for each indicator. Results across countries are presented for these indicators using the diagnostic criteria. Third, a rubric for identifying possible underlying causes for each outcome indicator is shown. This diagnosis can be used to identify an array of possible actions to improve food security, diet, health, and environmental outcomes. Two country case studies are presented to illustrate how this approach can be used to diagnose likely challenge areas in two different types of settings and point toward possible actions to address these challenges. This is the first paper to identify possible cutoffs to signal low to high likelihood challenge areas across a suite of key food systems indicators.

## Materials and methods

### Identification of diagnostic indicators

The FSD includes indicators relevant to the food systems conceptual framework from the Food Systems Countdown Initiative, which was adapted from the UN High-Level Panel of Experts on Food Systems and Nutrition report (Fig 1) [1, 29]. Not all the indicators available on the FSD (over 200) are useful in diagnosing challenges in achieving nutrition and environmental outcomes; many are purely descriptive without any causal relationship to outcomes (e.g., percent urban population). To select diagnostic indicators, the following criteria were applied: 1) the indicator has a clear target value or direction (i.e. higher is better, lower is better, or a certain range is better); 2) the target value is universal and not dependent upon context; 3) data for the indicator are available for the majority of countries; 4) data are recent (the indicator has been updated at least once since 2010, as older values may not be representative of the current status of a country); and 5) the indicator is globally acceptable and preferably available in the public domain.

A total of 39 diagnostic indicators were selected for the FSD diagnostic approach (Table 1). These indicators describe four major components of food systems illustrated in the conceptual framework (Fig 1): food supply chains; food environments; food security, diet, and nutrition outcomes; and environmental outcomes. All indicators and their sources are identified in Table 1. For food supply chains, five indicators were chosen that describe crop biodiversity and food losses. Production indicators, such as cereal and vegetable yield, were not included because appropriate thresholds for these indicators may depend on a country's agroecological setting. For the food environment, 11 indicators met the diagnostic criteria, encompassing food availability, food affordability, and product properties. For nutrition and food security outcomes, 14 indicators were selected that describe food security, diets, nutritional status for

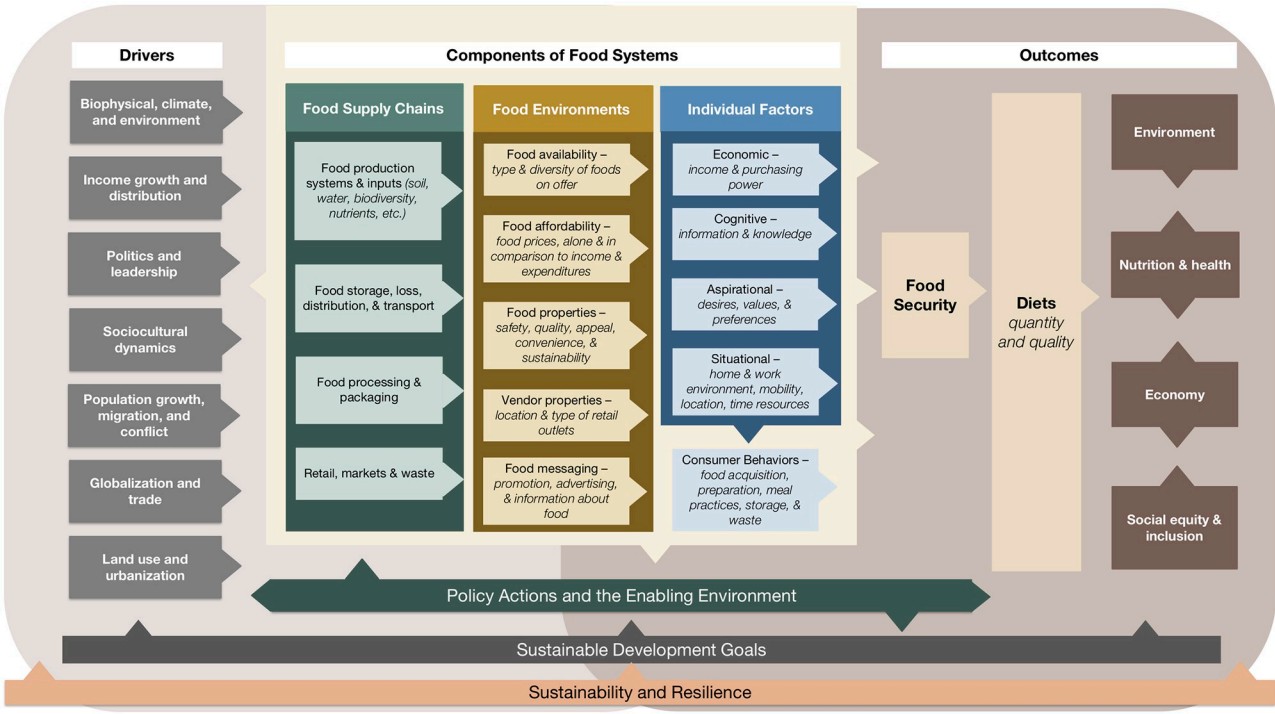

**Fig 1. Food systems framework.**

**Table 1. Selected indicators and cutoffs for food systems diagnostics.**

| Sector | Subsector | Indicator | Source | Year | # Countries | Unlikely Challenge Area Cutoffs (N) | Potential Challenge Area Cutoffs (N) | Likely Challenge Area Cutoffs (N) | Cutoff Type* |
|---|---|---|---|---|---|---|---|---|---|
| Food supply chains | Production systems and input supply | 1. Crop species richness (average number of crops/ unit of land) | IFPRI 2019 [34] | 2010 | 184 | >7 (87) | 3–7 (66) | <3 (31) | 2 |
| | Storage and distribution | 2. Cereal losses (% of domestic supply) | FAOSTAT [35] | 2018 | 156 | <2.5 (57) | 2.5–7 (77) | >7 (22) | 4 |
| | | 3. Pulse losses (% of domestic supply) | FAOSTAT [35] | 2018 | 150 | <2.5 (59) | 2.5–5 (64) | >5 (27) | 4 |
| | | 4. Fruit losses (% of domestic supply) | FAOSTAT [35] | 2018 | 166 | <5 (49) | 5–10 (89) | >10 (28) | 4 |
| | | 5. Vegetable losses (% of domestic supply) | FAOSTAT [35] | 2018 | 167 | <5 (35) | 5–10 (114) | >10 (18) | 3 |
| Food environment | Food availability | 6. Dietary energy in the food supply (kcal/capita/d) | FAOSTAT [35] | 2018 | 167 | ≥2500 (126) | n/a | <2500 (41) | 2 |
| | | 7. Dietary energy supply from cereals, roots, and tubers (%) | FAOSTAT [36] | 2016 | 168 | <40 (58) | 40–60 (75) | >60 (35) | 3 |
| | | 8. Fruit supply (g/capita/d) | FAOSTAT [35] | 2018 | 168 | >200 (85) | 100–200 (57) | <100 (26) | 2 |
| | | 9. Vegetable supply (g/capita/d) | FAOSTAT [35] | 2018 | 168 | >200 (88) | 100–200 (45) | <100 (35) | 2 |
| | | 10. Pulse supply (g/capita/d) | FAOSTAT [35] | 2018 | 168 | >60 (5) | 30–60 (24) | <30 (139) | 2 |
| | Product properties | 11. Retail value of UPFs (USD/capita/year) | Euromonitor [37] | 2018 | 188 | <100 (68) | 100–300 (60) | >300 (60) | 4 |
| | Food affordability | 12. Relative cost of adequate fruits and vegetables (ratio of the cost of the recommended amount of fruits and vegetables to the cost of the recommended amount of starchy staples per person per day) | Food Prices for Nutrition [38] | 2017 (est for 2018 and 2019) | 159 | <2 (20) | 2–4 (98) | >4 (41) | 3 |
| | | 13. Relative cost of adequate legumes, nuts, and seeds (ratio of the cost of the recommended amount of legumes, nuts, and seeds to the cost of the recommended amount of starchy staples per person per day) | Food Prices for Nutrition [38] | 2017 (est for 2018 and 2019) | 159 | <0.75 (94) | 0.75–1 (32) | >1 (33) | 2 |
| | | 14. Relative cost of healthy diet (ratio of the cost of a healthy diet to the cost of caloric adequacy) | Food Prices for Nutrition [38] | 2017 (est for 2018 and 2019) | 159 | <3.5 (38) | 3.5–5 (67) | >5 (54) | 3 |
| | | 15. Cost of an energy sufficient diet (2011 USD/capita/d) | Food Prices for Nutrition [38] | 2017 (est for 2018 and 2019) | 163 | <0.75 (74) | 0.75–1.20 (75) | >1.20 (14) | 2 |
| | | 16. Affordability of a healthy diet (ratio of the cost of a healthy diet to observed per capita food expenditures from national accounts) | Food Prices for Nutrition [38] | 2017 (est for 2018 and 2019) | 159 | <0.5 (59) | 0.5–1 (61) | >1 (38) | 2 |

(*Continued*)

**Table 1.** (Continued)

| Sector | Subsector | Indicator | Source | Year | # Countries | Unlikely Challenge Area Cutoffs (N) | Potential Challenge Area Cutoffs (N) | Likely Challenge Area Cutoffs (N) | Cutoff Type* |
|---|---|---|---|---|---|---|---|---|---|
| Food Security, Diets and Nutrition | Food security | 17. People who cannot afford a healthy diet (%) | Food Prices for Nutrition [38] | 2017 (est for 2018 and 2019) | 141 | <5 (45) | 5–25 (32) | >25 (64) | 4 |
| | | 18. Prevalence of moderate or severe food insecurity (%) (FIES) | FAOSTAT [36] | 2019 | 121 | <5 (13) | 5-25(52) | >25 (56) | 3 |
| | | 19. Prevalence of undernourishment (%) | FAOSTAT [36] | 2019 | 157 | <5 (72) | 5–10 (39) | >10 (46) | 4 |
| | Dietary intake | 20. Prevalence of minimum diet diversity (MDD) in infants age 6–23 months (%) | UNICEF [39] | 2013–2018 | 86 | >60 (11) | 30–60 (30) | <30 (45) | 3 |
| | | 21. Prevalence of infants (6–23 months) consuming zero fruits and vegetables (%) | UNICEF [39] | 2010–2020 | 97 | <25 (32) | 25–50 (40) | >50 (25) | 4 |
| | | 22. Prevalence of infants (6–23 months) consuming no meat, fish, or eggs (%) | UNICEF [39] | 2010–2020 | 97 | <30 (26) | 30–60 (48) | >60 (23) | 3 |
| | Nutritional status | 23. Prevalence of under-5 stunting (HAZ <-2 SD) (%) | UNICEF, WHO, and World Bank [40] | 2010–2019 | 125 | <10 (33) | 10–20 (27) | >20 (65) | 1 |
| | | 24. Prevalence of under-5 wasting (WHZ < -2 SD) (%) | UNICEF, WHO, and World Bank [40] | 2010–2019 | 124 | <5 (69) | 5–10 (39) | >10 (16) | 1 |
| | | 25. Prevalence of underweight in women (BMI <18.5 kg/m$^2$) (%) | NCD-RisC [41] | 2016 | 190 | <5 (123) | 5–10 (41) | >10 (26) | 1 |
| | | 26. Prevalence of anemia in women 15–49 years (%) | WHO Global Health Observatory [42] | 2016 | 187 | <20 (37) | 20–40 (115) | >40 (35) | 1 |
| | | 27. Prevalence of under-5 overweight and obesity (WHZ >2 SD) (%) | UNICEF, WHO, and World Bank [40] | 2010–2019 | 116 | <5 (53) | 5–10 (40) | >10 (23) | 1 |
| | | 28. Prevalence of adult obesity (BMI $\geq$ 30 kg/m$^2$) (%) | NCD-RisC [41] | 2016 | 190 | <10 (50) | 10–22.5 (56) | >22.5 (84) | 4 |
| | NCDs | 29. Prevalence of adult raised blood pressure (SBP $\geq$ 140 or DBP $\geq$ 90 mm Hg) (%) | NCD-RisC [43] | 2015 | 189 | <20 (36) | 20–25 (68) | >25 (85) | 3 |
| | | 30. Prevalence of diabetes (%) | NCD-RisC [44] | 2014 | 190 | <6 (27) | 6–10 (97) | >10 (66) | 3 |

(*Continued*)

**Table 1.** (Continued)

| Sector | Subsector | Indicator | Source | Year | # Countries | Unlikely Challenge Area Cutoffs (N) | Potential Challenge Area Cutoffs (N) | Likely Challenge Area Cutoffs (N) | Cutoff Type* |
|---|---|---|---|---|---|---|---|---|---|
| Environment Outcomes | Environment measures at consumption level | 31. GHGe of food consumption (kg $CO_2$-equivalent / capita) | WWF [45] | 2010 | 147 | <2000 (61) | 2000–2500 (28) | >2500 (58) | 4 |
| | | 32. Water use linked to food consumption (liters/capita) | WWF [45] | 2010 | 147 | <250 (49) | 250–350 (48) | >350 (50) | 3 |
| | | 33. Eutrophication of food consumption (g $PO_4$-equivalent /capita) | WWF [45] | 2010 | 147 | <7500 (48) | 7500–10000 (41) | >10000 (58) | 3 |
| | | 34. Biodiversity impact of food consumption (extinctions per species year*$10^{12}$/capita) | WWF [45] | 2010 | 147 | <350 (48) | 350–750 (47) | >750 (52) | 4 |
| | | 35. Total ecological footprint of consumption (global ha/ capita) | Global Footprint Network [46] | 2016 | 177 | <1.68 (55) | 1.68–2.75 (42) | >2.75 (80) | 2 |
| | Environment measures at production level | 36. Total ecological footprint of production (global hectares/ capita) | Global Footprint Network [46] | 2017 | 177 | <1.67 (77) | 1.67–2.75 (33) | >2.75 (67) | 2 |
| | | 37. Average number of threats to soil biodiversity | Orgiazzi et al. 2016 [47] | 1997–2010 | 181 | <1 (3) | 1–2 (101) | >2 (77) | 2 |
| | | 38. Agricultural land change from 2008 to 2018 (log(1,000 ha/ year)) | FAOSTAT [48] | 2018 | 193 | <0 (52) | 0–2 (39) | >2 (102) | 2 |
| | | 39. Average proportion of agricultural lands embedding at least 10% of natural vegetation (%) | Jones et al. 2021 [49] | 2015 | 234 | >50 (17) | 25–50 (65) | <25 (152) | 2 |

Cutoff type: 1) Published / pre-established cutoffs on prevalence of public health significance, 2) Cutoffs based on normative recommendations, 3) Cutoffs based on global distribution of data: Rounded tertiles based on normal distributions (see Figs 2 and 4) Cutoffs based on global distribution of data: Binning based on bimodal or skewed distributions (see Fig 2).

adults and children, and diet-related noncommunicable diseases (NCDs). Few diet indicators have been included due to lack of data, despite dietary outcomes being of high interest and importance as outcomes of the food system and being closely related to food environments as well as other nutrition, health, and environmental outcomes. The only measures of dietary intake included were three indicators of diet quality among infants and young children because they are the only diet quality indicators that are current and comparably collected across countries. These are collected by Demographic and Health Surveys (DHS) and are available mostly in low- and middle-income countries (LMICs). Dietary measures for other age groups (school-aged children, adults, and adolescents) do not currently meet the geographic distribution requirements to be included in the diagnostic approach, but diet quality data currently being collected by the Gallup World Poll and DHS will be added as soon as they are available, covering indicators of dietary adequacy and NCD risk factors in the general population [33]. For environmental outcomes, nine indicators met the diagnostic criteria and described production-level outcomes and consumption-level outcomes.

## Establishing cutoffs for each indicator

To establish cutoffs for each indicator, there was a need to develop criteria for flagging values that would indicate a likely challenge associated with each indicator. In many applications,

cutoffs are used to interpret continuous indicators, where a value on one side of the cutoff is diagnosed as problematic, while a value on the other side is diagnosed as acceptable. Because the severity of a condition is rarely tied to an exact value, but rather to a position of greater or lesser risk within a continuous range of values, setting cutoffs for diagnosis requires careful consideration. Each diagnostic indicator was categorized into three categories: green (unlikely challenge area), yellow (potential challenge area), or red (likely challenge area). Since different levels of evidence exist for each indicator, thresholds were established using four different methods, as follows. First, when possible, pre-defined cutoff values representative of global consensus on public health significance (such as pre-defined low to high categories for the prevalence of stunting in young children) were used (S1 Table). However, for most indicators, such pre-defined cutoff values do not currently exist. Second, where normative recommendations exist, these were used to establish cutoffs (S2 Table). For example, thresholds for fruit supply adequacy were based on globally recommended per capita intakes of fruit, with countries in the green category having a supply of fruit at or above the recommended intake and countries in the red category having a supply of less than half of the recommended amount. Third, where no cutoffs have been published and no normative values exist, the relative values of country data points can be compared as relatively higher or lower. For each indicator, density plots, a variation of histograms, were used to examine the distribution of data, using the data assembled on the FSD (S3 Table). A density plot was chosen over a histogram to view a smoothed distribution of the data using kernel density estimation. Most indicators had an approximately normal distribution and were divided into tertiles, rounded to interpretable values. We prioritized retaining meaningful or more easily interpretable cutoffs over exact tertiles. Fourth, some indicators had a bimodal or highly skewed distribution; in these cases, the peaks were bifurcated by the two cutoff points (low/medium; medium/high). An example of each of these is shown in Fig 2. The cutoffs for each indicator, as well as the method used to set them, are shown in Table 1.

Four example indicators are explained to demonstrate the methodology for determining the cut-offs. As mentioned above, the prevalence of stunting is an example of an indicator where cutoffs are based on published consensus on cutoffs [50]. An example of an indicator where cutoffs are based on normative recommendations is vegetable supply. This indicator is included as vegetable supply is a precursor of vegetable consumption; thus, the cutoffs are set based on the World Health Organization's recommendation for vegetable consumption as part of a healthy diet. Vegetable losses, on the other hand, is an example of an indicator where no normative cutoffs or recommendations exist. Because the data for this indicator are normally distributed across countries, the cutoffs are set using rounded tertiles. The prevalence of adult obesity similarly has no published or accepted cutoffs for public health significance, but the distribution shows two large peaks, so bimodal curve-based binning is used to set cutoffs.

## Analysis of food systems diagnosis across countries

The analysis of national-level data included 195 countries globally. The most recent data available for all countries was used. Countries for which the most recent value was prior to 2010 were excluded. For visualization and analysis, countries were stratified by the 2022 World Bank income classification [51]. Analysis, visualization, and data management were conducted using the R Statistical Computing Environment (version: 3.6.2) [52].

## Identifying actions for addressing challenge areas

Diagnosing challenging areas across food systems begs the question, "then what?" The intention of the diagnostic approach is to spur policy debate and advocacy for possible solutions to the challenge areas. To aid this process, a menu of possible actions can be linked to each

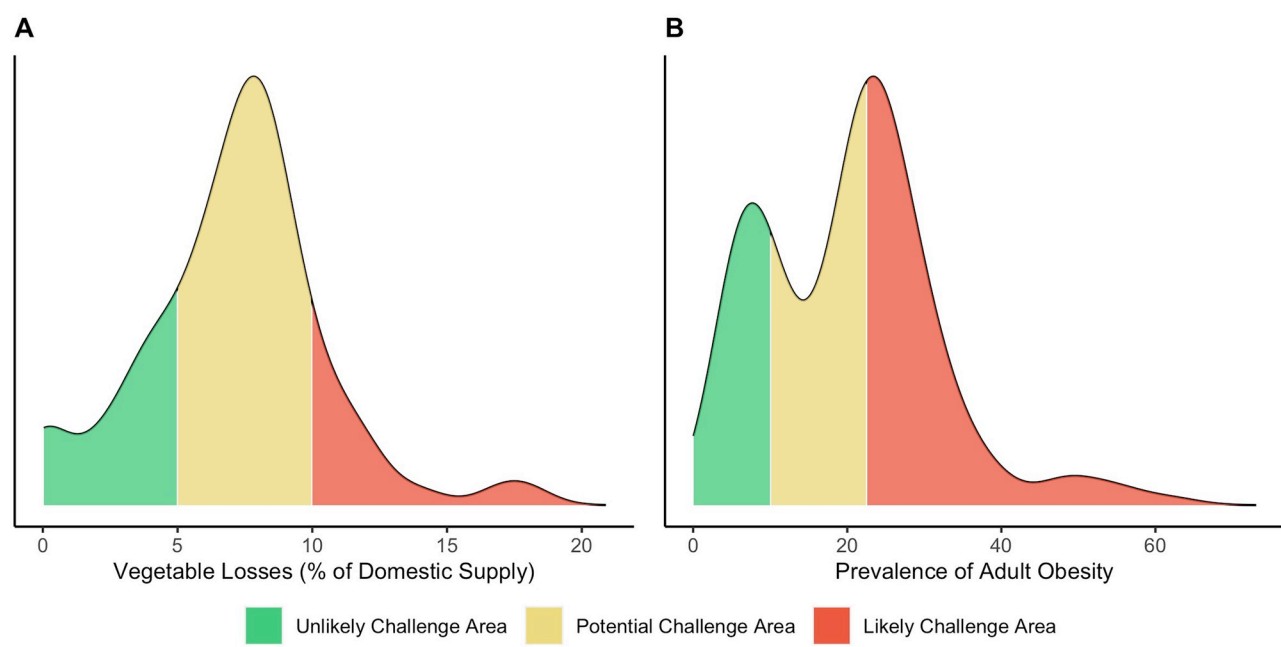

**Fig 2. Examples of diagnose thresholds for A) unimodal and B) bimodal indicator frequency distributions.** Density plots of the data distribution for A) vegetable losses (% of domestic supply) and B) prevalence of adult obesity with colors indicating cutoffs for diagnostic thresholds: green indicates an unlikely challenge area, yellow indicates a potential challenge area, and red indicates a likely challenge area. Density plots are similar to histograms but use kernel density smoothing, rather than bins, to present a continuous distribution of the data. The peaks of each plot represent where the highest number of observations exists in the data.

challenge area. While possible actions are primarily up to the users to deliberate and decide, and may be very context specific, the diagnostic approach provides evidence to inform this deliberation, and a selection of possible evidence-based policies and actions to consider toward improving outcomes for each challenge [53]. Each of the diagnostic indicators is matched with other indicators in the FSD (Table 2), providing a road map to other potential contributing factors upstream that may provide deeper understanding into the causal pathway. Some outcomes have multiple food and non-food causes (e.g., poor nutritional status); only the possible causes related to food (e.g., food insecurity and inadequate diets) are identified.

## Case studies

To demonstrate the use of the diagnostic approach in specific settings, two country case studies are presented. Tanzania and Greece were chosen to demonstrate how the diagnostic approach can be applied to different types of food systems, Tanzania having a predominantly rural and traditional food system and Greece an industrial and consolidated food system [54]. Furthermore, diet quality data for the general population were available from these two countries, which allowed for a richer analysis of the problems that food systems may need to address. Comparable diet quality data are currently being collected by the Gallup World Poll and DHS and will soon be available for a growing number of countries [33].

## Results

### Applying the diagnostics to national food systems

Of the 195 countries assessed in the analysis, the average country coverage for indicators was 158 or 81% of countries (Table 1). Five indicators had established prevalence thresholds for

**Table 2. Diagnose indicators linked to potential contributing indicators where data are available in the food systems dashboard.**

| Diagnostic Indicator | Potential Contributing Indicators |
|---|---|
| 1. Crop species richness | |
| 2. Cereal losses | agricultural infrastructure index |
| 3. Pulse losses | agricultural infrastructure index |
| 4. Fruit losses | agricultural infrastructure index |
| 5. Vegetable losses | agricultural infrastructure index |
| 6. Dietary energy in food supply | cereal import dependency ratio, cereal yield |
| 7. Dietary energy from cereals, roots, and tubers | supply of vegetables, fruit, pulses, milk, meat, fish, and eggs; relative cost of adequate fruits and vegetables; relative cost of adequate legumes, nuts, and seeds, relative caloric prices (RCPs) |
| 8. Fruit supply | fruit losses |
| 9. Vegetable supply | vegetable yield, vegetable losses |
| 10. Pulses supply | pulse losses |
| 11. Retail value of UPFs | existence of any policies on marketing of junk food to children |
| 12. Relative cost of adequate fruits and vegetables | fruit supply; vegetable supply; dietary energy from cereals, roots, and tubers |
| 13. Relative cost of adequate legumes, nuts, and seeds | pulses supply; dietary energy from cereals, roots, and tubers |
| 14. Relative cost of healthy diet to cost of caloric adequacy | relative cost of adequate fruits and vegetables; relative cost of adequate legumes, nuts, and seeds, |
| 15. Cost of an energy sufficient diet | dietary energy in food supply, cereal losses |
| 16. Affordability of a healthy diet (the ratio of the cost of a healthy diet to observed per capita food expenditures from national accounts) | relative cost of adequate fruits and vegetables; relative cost of adequate legumes, nuts, and seeds; RCPs; consumption expenditures |
| 17. People who cannot afford a healthy diet | relative cost of a healthy diet, cost of a healthy diet relative to food expenditures, socioeconomic drivers |
| 18. Prevalence of moderate or severe food insecurity (%) (FIES) | dietary energy in the food supply, socioeconomic drivers |
| 19. Prevalence of undernourishment (%) | dietary energy in the food supply, socioeconomic drivers |
| 20. Prevalence of minimum diet diversity | dietary energy from cereals, roots, and tubers; availability of each food group; relative cost of a healthy diet; affordability of a healthy diet; socioeconomic drivers |
| 21. Prevalence of infants (6–23 months) consuming zero fruits and vegetables (%) | dietary energy from cereals, roots, and tubers; availability of each food group; relative cost of a healthy diet; affordability of a healthy diet; socioeconomic drivers |
| 22. Prevalence of infants (6–23 months) consuming no meat, fish, or eggs (%) | dietary energy from cereals, roots, and tubers; availability of each food group; relative cost of a healthy diet; affordability of a healthy diet; socioeconomic drivers |
| 23. Prevalence of under-5 stunting | infant and young child feeding (IYCF) indicators; relative cost of a healthy diet; affordability of a healthy diet; dietary energy from cereals, roots, and tubers; socioeconomic drivers |
| 24. Prevalence of under-5 wasting | dietary energy in the food supply, IYCF indicators, socioeconomic drivers |
| 25. Prevalence of underweight in women | dietary energy in the food supply, socioeconomic drivers |

(*Continued*)

**Table 2.** (Continued)

| Diagnostic Indicator | Potential Contributing Indicators |
|---|---|
| 26. Prevalence of anemia in women | supply of vegetables, pulses, and meat; dietary energy from cereals, roots, and tubers; relative cost of a healthy diet; affordability of a healthy diet |
| 27. Prevalence of under-5 overweight and obesity | dietary energy in the food supply, relative cost of healthy diet, affordability of a healthy diet, RCPs, retail share of UPFs, supply of sugar and oil |
| 28. Prevalence of adult obesity | dietary energy in the food supply, relative cost of healthy diet, affordability of a healthy diet, RCPs, retail share of UPFs, supply of sugar and oil |
| 29. Prevalence of adult raised blood pressure | dietary energy in the food supply, relative cost of a healthy diet, affordability of a healthy diet, RCPs, retail value of UPFs, supply of vegetables and fruit, supply of sugar and oil |
| 30. Prevalence of diabetes | dietary energy in the food supply, relative cost of a healthy diet, affordability of a healthy diet, RCPs, retail value of UPFs, taxes on sugar-sweetened beverages (SSBs), supply of vegetables and fruit, supply of sugar and oil |
| 31. GHGe of food consumption | dietary intake indicators, especially red meat and dairy |
| 32. Water use linked to food consumption | dietary intake indicators, especially red meat and dairy |
| 33. Eutrophication of food consumption | fertilizer consumption, nutrient nitrogen per ha of arable land, nutrient phosphate per ha of arable land, dietary intake indicators, especially red meat and dairy |
| 34. Biodiversity impact of food consumption | percent of intact area, agricultural land change |
| 35. Total ecological footprint of consumption (global ha/capita) | dietary intake indicators, especially red meat and dairy |
| 36. Total ecological footprint of production | crop species richness, agricultural land change, GHGe from agriculture |
| 37. Average number of threats to soil biodiversity | agricultural land as percentage of country land, nutrient nitrogen per ha of arable land, nutrient phosphate per ha of arable land, per capita biodiversity impact of food consumption, per capita eutrophication of food consumption |
| 38. Agricultural land change from 2008–2018 | percent of intact area, agricultural land as percentage of country land |
| 39. Average proportion of agricultural lands embedding at least 10% of natural vegetation | agricultural land as percentage of country land, agricultural land change |

public health significance: prevalence of wasting in children under 5 years (WHZ < -2), prevalence of stunting in children under 5 years (HAZ <-2), prevalence of underweight in women (BMI <18.5), prevalence of anemia in women 15–49 years, and prevalence of overweight and obesity in children under 5 years (WHZ >2). For 13 indicators, cutoffs were based on global recommendations, and for the remaining 21 indicators, cutoffs were based on the global distribution of the data (Table 1).

Taking a systems approach, Figs 3 and 4 bring the indicators together, highlighting patterns of challenge areas across the set of 39 indicators. Fig 3 shows the percentage of countries that have a likely challenge area for each indicator by country income classification [51]. Patterns in likely challenge areas are visible by income status, with some indicators moving more or less strongly with income, or in different directions. For example, supply of dietary energy and of fruits and vegetables are frequently flagged as likely challenge areas in lower-middle-income countries, but not often in upper-middle- or high-income countries. Meanwhile, pulse supply appears to be low across all income groups, though the relative cost of legumes is particularly a

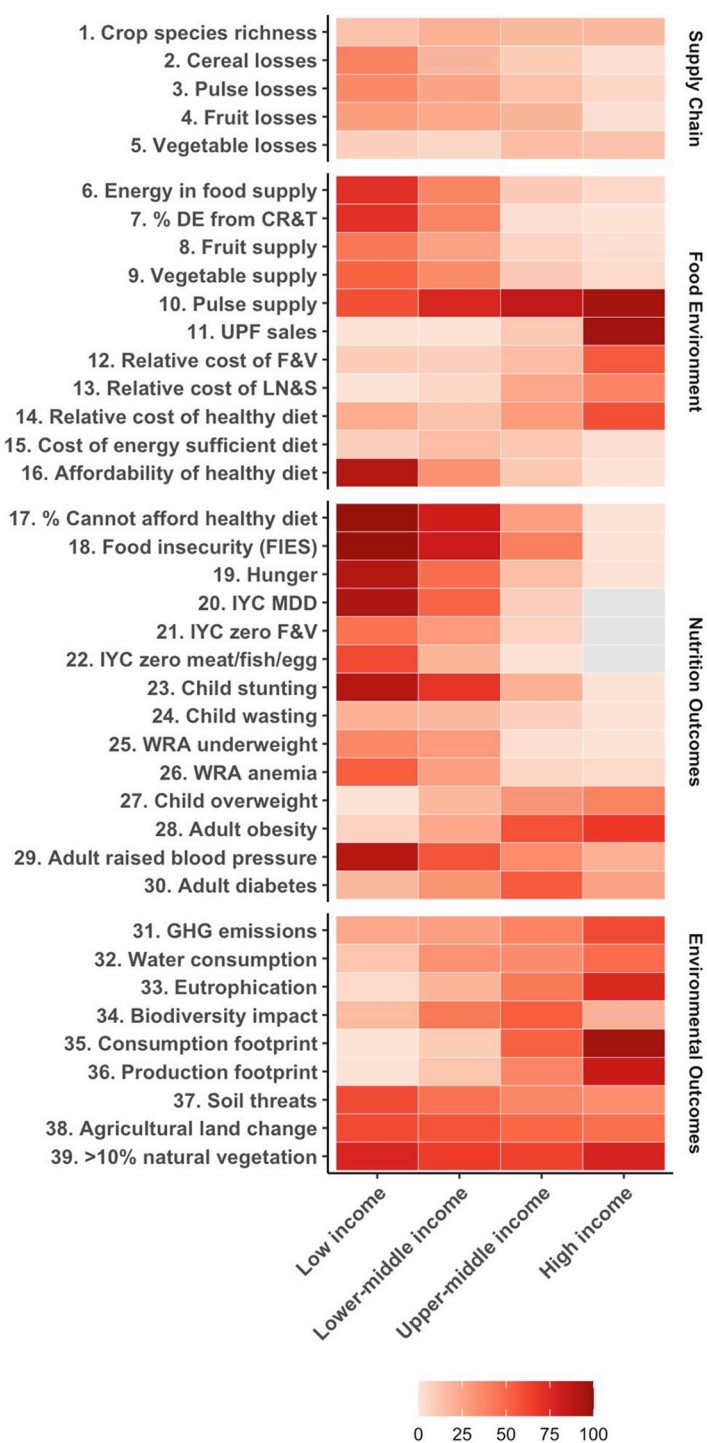

**Fig 3. Percentage of countries with likely challenge areas by income status.** The color indicates the percentage of countries facing likely challenge areas. Grey indicates <5 countries within an income group have data for the indicator. Full indicator names are listed in Table 1. DE: dietary energy; CR&T: cereals, roots, and tubers; UPF: ultra-processed foods; F&V: fruits and vegetables; LN&S: legumes, nuts, and seeds; IYC: infant and young child; MDD: minimum dietary diversity; WRA: women of reproductive age, GHG: greenhouse gases.

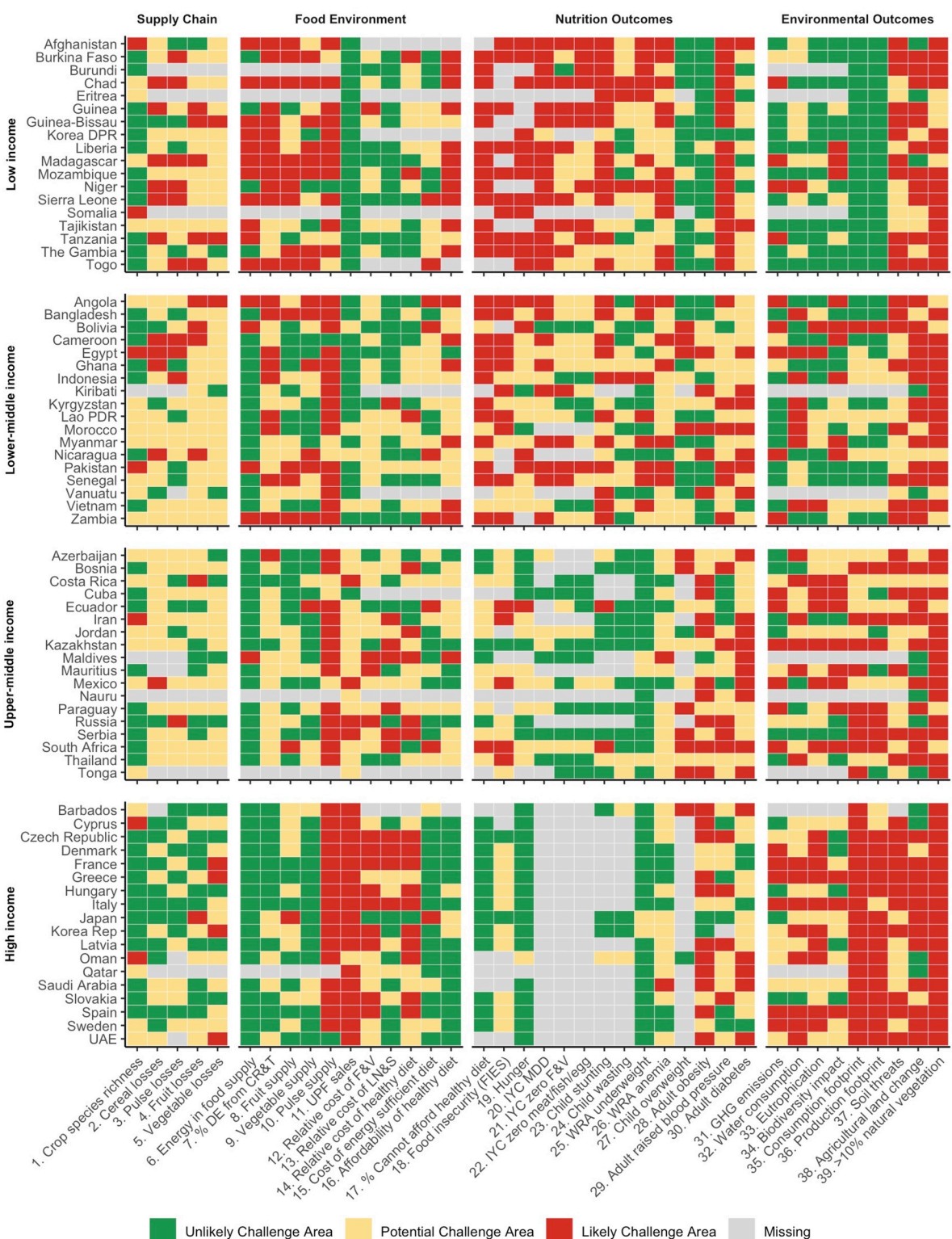

**Fig 4. Food systems diagnosis by country.** 18 countries were randomly selected from each income group to display food systems diagnosis patterns across income group. Full indicator names are listed in Table 1. DE: dietary energy; CR&T: cereals, roots, and tubers; UPF: ultra-processed foods; F&V: fruits and vegetables; LN&S: legumes, nuts & seeds; IYC: infant and young child; MDD: minimum dietary diversity; WRA: women of reproductive age, GHG: greenhouse gases.

challenge in higher-income settings. The percentage of the population who are hungry, food insecure, or who cannot afford a healthy diet are challenges in low-income countries, reflected in the dietary outcomes of low dietary diversity and low consumption of fruits, vegetables, and animal source foods among infants and young children in low-income countries. Sales of UPFs and adult obesity are challenges particularly in high-income countries. The set of nutrition outcome indicators tend to show nutrition transitions that mirror the food environment and dietary patterns. While low-income countries are mainly grappling with child undernutrition and food insecurity and high-income countries are largely grappling with adult obesity [55], middle-income countries are dealing with double burdens of malnutrition challenges [56]. Notably, however, adult raised blood pressure is much more problematic the lower the income, despite being an indicator of NCD risk. Moreover, diabetes presents the most significant challenge in upper-middle-income countries, not high-income countries. On the environmental side, eutrophication, GHGe, and consumption footprints are particular challenge areas in high-income countries, while threats to soil biodiversity, agricultural land change, and natural vegetation within agricultural landscapes are pressing challenge areas across countries of all incomes.

Each country faces a unique set of likely challenge areas across the food system or within a subsector of the food system. Fig 4 shows the diversity of country-level challenges within a randomly selected set of countries in each income classification. There are many countries which follow typical patterns seen by income classification, including greater challenge areas of undernutrition in low- and middle-income countries (e.g., anemia) and greater challenge areas of obesity and UPF sales in high-income countries. But there are also interesting country outliers for many indicators. For example, child wasting is an unlikely challenge area for several low-income countries, including Tanzania, Mozambique, and Liberia; UPF sales are atypically high in Costa Rica, Mexico, Russia, and Serbia compared to other low- and middle-income countries; and the low affordability of a healthy diet stands out in the Maldives. On the environmental side, the food supply chains of the Gambia, Liberia, and Mozambique have fewer challenge areas compared to other low-income countries. Few food supply chain indicators are flagged as challenges in high-income countries, but there are some notable exceptions on food losses in individual countries, such as high fruit losses in Japan and high vegetable losses in Greece and Korea. Positive deviants can also be identified. For example, Cyprus and Japan have relatively fewer food systems-related environmental challenge areas than other high-income countries.

Performance across indicators within a specific food systems component, within an individual country, is typically varied, rarely consisting of all likely challenge areas or no likely challenge areas. For example, Angola, a lower-middle-income country, has several likely challenge areas in the food environment related to the availability of food—including the supply of vegetables, pulses, and the overall dietary energy supply—and the cost of an energy sufficient diet is also a likely challenge. However, the premium consumers must pay for nutrient-dense foods, evident in the relative cost of fruits, vegetables, and pulses, and the relative cost of a healthy diet, is not a likely challenge area, as it is in many higher-income countries. Still, the cost of a healthy diet relative to household food expenditure (affordability) is a likely challenge area, which may indicate that the general cost of food, across all food groups, is still high.

To use the diagnosis to inform decision-making, one of the first steps is to explore the possible factors related to each challenge area. In Table 2, such factors are identified among indicators where data are available on the FSD, following the food systems conceptual framework (Fig 1). For example, the high prevalence of infants and young children with zero fruit and vegetable intakes might trace back to high cost of fruits and vegetables, and in turn low availability of fruits and vegetables, possibly linked to the supply chain issues of low crop

biodiversity and/or high fruit and vegetable losses. Countries that have high unaffordability of healthy diets tend to have low supply of fruits and vegetables.

### Applying the diagnostics in two country case studies

**Tanzania.** Tanzania is a low-income country with a food system that is predominantly rural and traditional [54]. The country has made steady progress in combating child stunting, which fell by approximately 10% from 2010 to 2018 [40]. However, 32% of children under five are stunted today—well above the 20% prevalence cutoff indicating a likely challenge area—and progress towards the elimination of stunting, a target within SDG 2, remains an unfinished agenda [57]. Though stunting is a multisectoral challenge with determinants beyond the food system, the diagnostic approach can help identify priority areas to be addressed in order to maximize the food system's contribution to ending stunting.

The FSD shows that Tanzania performs relatively well on breastfeeding, with nearly 60% of infants exclusively breastfed for the first six months of life and 92% still breastfed at one year, but complementary feeding still requires more attention [53]. Just 21% of children 6–23 months of age achieve minimum dietary diversity (MDD), making this a likely challenge area for Tanzania, and a probable cause of stunting. Unpacking MDD further, just 35% of children 6–23 months of age consume any meat, eggs, or fish, making this a likely challenge area, while consumption of fruits and vegetables are a potential challenge area with 29% consuming zero fruits and vegetables in the previous day [39]. Animal-source foods (ASF) are important for child growth, due to their favorable amino acid profile and their high density of micronutrients such as iron and zinc [58, 59].

The diagnostic approach can be used to trace further causal pathways through other areas of the food environment and food supply chains. Particularly relevant for MDD are the availability and affordability of diverse foods. Fifty-six percent of Tanzania's dietary energy supply is derived from cereals, roots, and tubers, which is a potential challenge area. The affordability of a healthy diet may be another area of concern, also flagged as a potential challenge area, though relative costs of fruits, vegetables, and pulses are low.

Recognizing the intergenerational nature of stunting, examining women's nutritional status and dietary intake may also shed light on possible causes of stunting. Nutritional status at the preconception stage and during pregnancy may influence intrauterine growth and birth outcomes [60]. The diagnostic approach indicates that anemia—which has both dietary and non-dietary causes—is a significant problem in Tanzania, affecting 37% of women of reproductive age. Diet Quality Questionnaires (DQQ) collected in Tanzania from the Global Diet Quality Project provide more insights, including that only 63% of women consumed an ASF during the previous day compared with 71% of men. ASF consumption has been associated with reducing the risk for small-for-gestational age and low birthweight babies [61, 62]. Looking at the sociocultural drivers of the food system, Tanzania's gender inequality index is high, which is consistent with this gender disparity in diets.

After identifying likely challenge areas that may be worth more in-depth, contextualized analysis, national stakeholders may be a step closer to selecting policies and actions that may be appropriate to address these challenges. In this example related to stunting in Tanzania, these could include investing in market infrastructure to enhance access to nutritious food and utilizing social protection platforms to enhance the purchasing power of women, especially around pregnancy.

**Greece.** Greece is a high-income country and its food system is indicative of an industrial and consolidated typology [54]. Countries associated with the Mediterranean Diet, like Greece, have historically consumed diets that are low in red meat and high in plant foods, including

pulses, with high fat intake from olive oil [63, 64]. Greece has 747 grams of fruits and vegetables available per person per day, an abundant supply making it likely that most people in Greece would be able to access at least 400 grams of fruits and vegetables per day, the WHO-defined minimum [65]. However, Greece's national pulse supply is just 14 grams per person per day, indicating a likely challenge area, while other Mediterranean countries, including Italy and Spain, are 14 and 15 grams, respectively, and France is just 4.7 grams per person per day, indicating it is a likely challenge area for all of these countries. As this diagnostic exercise demonstrates in Fig 4, a common challenge for many countries is to provide sufficient supply of pulses in their food environments, but this is especially problematic for high-income countries. Pulses could play a key role in transforming food systems for improved nutrition and environmental sustainability, as they are less intensive in their GHGe and use of water than other protein-rich foods, and their consumption has been associated with reductions in key NCD-related risk factors, including low-density lipoprotein (LDL) cholesterol concentration and blood pressure [6].

Recognizing the influence food environments have on consumer behavior and ultimately diet quality, a next step in this analysis might be to investigate whether diets are, in fact, also low in pulses. DQQ data from the Global Diet Quality Project indicate that in Greece, pulses are indeed a dietary gap, with just 18% of a nationally representative sample having consumed pulses in the day prior to the survey; this is coupled with relatively high consumption of red meat (44%) and processed meat (23%), and in contrast to high consumption of fruits and vegetables (95%) [33]. These diet data indicate that higher pulse consumption could substitute for some red and processed meat consumption, with co-benefits for NCD risk and environmental impact. In addition to the low physical supply, low pulse consumption could be brought on by unaffordability of pulses; however, in Greece the cost of pulses relative to starchy foods is cheap, indicating that cost is less likely to be a contributor.

Examining its production-related indicators, Greece performs well on crop species richness, but has a likely challenge area related to average threats to soil biodiversity. Greece's average soil organic matter is also 47 tonnes per hectare, slightly lower than the Southern Europe regional average of 59 tonnes per hectare [66].

A policy area for consideration to address these likely challenge areas may be to realign agricultural incentives towards increased production of pulses. Greater integration of pulses in agriculture may present an opportunity to improve environmental outcomes. Agroecological approaches emphasize agrobiodiversity as a means of enhancing the natural resources and ecosystem services that support sustainable yield gains, with low environmental impacts [67]. Inclusion of pulses in intercropping, cover cropping, and crop rotation strategies has been shown to improve soil structure, nitrogen fixing, and pest management [68–70].

These factors suggest that pulses could feature well in a dual strategy to shift diets and improve soil quality in Greece. Agriculture policy could incentivize pulse production to increase availability and environmental co-benefits. Consumer demand creation activities centered around the Mediterranean diet could also be considered to complement agriculture policy that includes or focuses on pulses.

## Discussion

This paper is the first of its kind to develop a methodology to diagnose food systems' performance to help inform food systems governance and accountability. The results indicate certain clear and consistent trends across income groups. However, each country faces a unique set of likely challenge areas. While many trends observed by income classification may be intuitive, the diagnostic approach presented here adds numbers and nuances to these trends and

supports the consideration of multiple likely challenge areas together. Jointly, this approach suggests a high potential for learning from different policy and programmatic interventions across countries–e.g., by identifying the positive deviants for a given indicator within a particular income classification or food system type, by connecting challenge areas, and by understanding the reasons behind successes and which ones could be replicated in other contexts.

As illustrated by the above case studies, this diagnostic approach can inform policy making. For countries where the diagnosis suggests unlikely challenge areas, policies can be encouraged to sustain success and share lessons learned. For likely challenge areas, policies can be encouraged to improve the highlighted sub-optimal outcomes. The diagnostic approach also helps identify bundles of challenge areas for policy action: for each nutrition outcome, a road map is provided to relevant indicators within the food supply chain and food environment. Diagnosis within these food supply and food environment indicators pinpoints areas of relatively poor performance upstream from diet outcomes, where attention can be focused on context-specific policy actions that could improve outcomes. In other words, the diagnostic approach identifies both the symptoms of a malfunctioning food system as well as potential contributing factors, providing evidence to then suggest an appropriate set of interventions or treatments to consider. This analysis will be further strengthened in future iterations of the FSD with additional dynamic tools that can use data to guide decision-making.

It is important to note that the diagnostic approach uses indicators to highlight likely challenge areas within food systems, but for many indicators the cutoffs were selected based on countries' relative performance, rather than absolute standards or targets. In addition, the indicators themselves are rarely an addressable problem–and should not be viewed as such. Rather, each indicator highlights one outcome of a complex causal chain of actions and interactions, along which there are several potential intervention points. For example, child stunting is a useful marker of delayed development and later chronic disease risk and indicative of multiple forms of deprivation occurring over a period of time–e.g., suboptimal nutrition, inadequate care, regular infection [71]. From a policy perspective, the key concerns are the underlying determinants and associated developmental outcomes of stunting. A high level of stunting indicates multiple underlying problems and should lead policy makers to seek to address these determinants (and their determinants). A proper diagnosis can thus begin with the indicator but not end there–instead looking for the possible points of leverage along the causal chain to that outcome. These points of leverage will vary across contexts and need to be interpreted with that local insight. Other indicators available on the FSD and elsewhere can help with this analysis–as indicated in the case studies shown above–but will also need to be combined with qualitative knowledge about the local culture, political economy, and which actions are likely to be most impactful. It is thus a guiding tool–not a determinative algorithm.

Previous efforts have developed aggregate indices to assess food systems sustainability and performance [72, 73]. Indices developed by Béné et al. and Chaudhary et al. encompass 25 to 27 indicators, respectively, which are used to calculate a composite score. Indicators and composite metrics used to describe food systems in these two papers are continuous, which is useful to avoid misclassification, but from a policy standpoint, it is harder to identify areas within the food system for policymakers and other stakeholders to intervene. To our knowledge, the present paper is the first attempt to undertake a systematic food systems diagnosis using a dashboard approach with a diverse set of indicators spanning food systems components and applying this across countries.

Strengths of this work include the use of a food systems framework (Fig 1) [29] to guide the identification of priority indicators and their interpretation, leveraging a uniquely broad dataset (both in terms of geographical coverage and food systems components) from the FSD. It is also highly transparent, with all data publicly available and all thresholds and approaches for

setting them presented here. The relative simplicity of the approach, which leverages the best available data and evidence from diverse sources but translates this into an easily understood 'stoplight' rating, is also an advantage, although it comes at a cost of masking complexity. When considering use for policy, this simplification is useful, as excess complexity can be paralyzing and difficult for non-specialists to interpret. The work has also helped to advance understanding on development of *actionable* food systems indicators–that is, highlighting which indicators (among a large number available) can be used to inform real-world decisions.

There are also certain limitations to this work. First, narrowing focus to just a few dozen indicators was necessary to prioritize and make the diagnostic approach understandable and actionable, but it may leave out other indicators that are also meaningful, especially in specific country contexts. In addition, there are certain components and outcome areas of the food system, such as livelihoods and cultural identity, which are not well covered with high-quality, relevant indicators–and are thus necessarily excluded here. Dietary data are also an important gap: due to limited availability of robust dietary data for most countries, dietary outcomes (aside from MDD, prevalence of infants 6–23 months consuming zero fruit or vegetables, and prevalence of infants 6–23 months consuming no meat, fish, or eggs) are omitted until they become available across countries. In the future, the FSD will include more dietary outcomes to better assess diets as the critical link between food environments and nutrition and environmental outcomes. These outcomes will include the minimum dietary diversity for women of reproductive age (MDD-W); an indicator of consumption of all five recommended food groups (vegetables; fruits; pulses, nuts, and seeds; animal source foods; and starchy staples); and indicators of risk factors for NCDs defined within WHO and other global recommendations, including consumption of adequate fruits and vegetables; whole grains; pulses, nuts, and seeds; and fiber and limited consumption of free sugar, salt, fat, saturated fat, and red and processed meat [33]. It is also recognized that the quality of data for certain indicators (e.g., GHGe) might differ between countries and that might affect identified patterns. Second, this systems approach allows users to consider bundles of challenge areas and draw potential connections between those, but to make statements about causality, more in-depth analysis is needed. Third, the presented results focus at the global and national levels and do not consider subnational data–even though certain countries (e.g., India) have considerable subnational diversity within their food systems as well as locally devolved policymaking processes. Fourth, many of the indicators come from official global repositories, the most reliable and comparable data sources (e.g., FAOSTAT); however, these often poorly capture the role of wild or local foods in diets, the environment, and local economies [49]. Finally, for indicators where no cutoffs have been published and no normative values exist, the cutoffs are based on density plots and countries' relative performance. These cutoffs could be refined in the future with more evidence of meaningful normative values.

There are several opportunities to build on this work. First, identifying potential challenge areas through this quantitative approach can trigger and support in-depth context-specific analysis, which includes stakeholder consultation and the integration of qualitative information to provide a more nuanced diagnosis and resulting decision options. National stakeholders may also enrich their analyses by supplementing the diagnosis with other data available at country-level, as has been demonstrated in the case studies in their drawing on DQQ data for Tanzania and Greece. Second, each of the diagnostic indicators could be paired with relevant policy and programmatic innovations (be they technological, nature-based, or societal) to improve both diets and planetary health. While no single action can fix food systems, governments, non-governmental organizations, civil society, and businesses can each act to start to transform food systems. It is hoped that the diagnostics presented in this paper are a step

towards better monitoring of food systems performance that can lead to stronger governance and accountability of food systems and their transformation.

## Supporting information

**S1 Table. Diagnosis indicators where thresholds were established using public health guidance (Cutoff Type 1).**
(DOCX)

**S2 Table. Diagnosis indicators where thresholds were established using public health and environmental recommendations (Cutoff Type 2).**
(DOCX)

**S3 Table. Diagnosis indicators where thresholds were established by consulting histogram of global distribution (Cutoff Types: 3 (unimodal) & 4 (bimodal or skewed)).**
(DOCX)

## Author Contributions

**Conceptualization:** Anna Herforth, Alexandra L. Bellows, Quinn Marshall, Rebecca McLaren, Ty Beal, Stella Nordhagen, Roseline Remans, Jessica Fanzo.

**Data curation:** Anna Herforth, Alexandra L. Bellows, Quinn Marshall, Ty Beal, Roseline Remans, Natalia Estrada Carmona.

**Formal analysis:** Anna Herforth, Alexandra L. Bellows, Ty Beal, Roseline Remans.

**Project administration:** Rebecca McLaren.

**Supervision:** Jessica Fanzo.

**Visualization:** Alexandra L. Bellows.

**Writing – original draft:** Anna Herforth, Alexandra L. Bellows, Quinn Marshall, Rebecca McLaren, Ty Beal, Stella Nordhagen, Roseline Remans, Jessica Fanzo.

**Writing – review & editing:** Anna Herforth, Alexandra L. Bellows, Quinn Marshall, Rebecca McLaren, Ty Beal, Stella Nordhagen, Roseline Remans, Natalia Estrada Carmona, Jessica Fanzo.

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
