## [Decision Letter · Decision Letter 0]

10 May 2022

PONE-D-22-10677Diagnosing the performance of food systems to increase accountability towards healthy diets and environmental sustainabilityPLOS ONE

Dear Dr. McLaren,

Thank you for submitting your manuscript to PLOS ONE. After careful consideration, we feel that it has merit but does not fully meet PLOS ONE’s publication criteria as it currently stands. Therefore, we invite you to submit a revised version of the manuscript that addresses the points raised during the review process.

We look forward to receiving your revised manuscript.

Kind regards,

Fatih Oz, Ph.D.

Academic Editor

PLOS ONE

Journal Requirements:

When submitting your revision, we need you to address these additional requirements. 1. Please ensure that your manuscript meets PLOS ONE's style requirements, including those for file naming. The PLOS ONE style templates can be found at https://journals.plos.org/plosone/s/file?id=wjVg/PLOSOne_formatting_sample_main_body.pdf and https://journals.plos.org/plosone/s/file?id=ba62/PLOSOne_formatting_sample_title_authors_affiliations.pdf
 2. Please review your reference list to ensure that it is complete and correct. If you have cited papers that have been retracted, please include the rationale for doing so in the manuscript text, or remove these references and replace them with relevant current references. Any changes to the reference list should be mentioned in the rebuttal letter that accompanies your revised manuscript. If you need to cite a retracted article, indicate the article’s retracted status in the References list and also include a citation and full reference for the retraction notice.

Additional Editor Comments:

Dear McLaren,

Thank you for submitting your manuscript to Plos One. I have completed my evaluation of your manuscript. As appended below, the reviewers have raised minor concerns/critiques and suggested further justification/work to consolidate the findings. Do go through the comments and amend the manuscript accordingly.

Reviewers' comments:

Reviewer's Responses to Questions

**Comments to the Author**

1. Is the manuscript technically sound, and do the data support the conclusions?

Reviewer #1: Yes

Reviewer #2: Yes

Reviewer #3: Yes

2. Has the statistical analysis been performed appropriately and rigorously? 

Reviewer #1: Yes

Reviewer #2: Yes

Reviewer #3: Yes

3. Have the authors made all data underlying the findings in their manuscript fully available?

Reviewer #1: Yes

Reviewer #2: Yes

Reviewer #3: Yes

4. Is the manuscript presented in an intelligible fashion and written in standard English?

Reviewer #1: Yes

Reviewer #2: Yes

Reviewer #3: Yes

5. Review Comments to the Author

Reviewer #1: The manuscript entitled “Diagnosing the performance of food systems to increase accountability towards healthy diets and environmental sustainability” presents a diagnostic methodology for 39 indicators representing food supply, food environments, nutrition outcomes, and environmental outcomes that offer cutoffs to assess performance of national food systems. The topic and results of study are interesting and the study presents valuable information that can be used to generate actions and decisions for healthy, safe, accessible, affordable and sustainable diets in order to improve human and planetary health. However there is a couple of issue to be answered by authors. My main comments on this manuscript are provided below:

-As known, generally agreed healthy diet constitutes are adequate amounts of fruit and vegetables, whole grains, legumes and nuts besides sufficient but not excessive intake of kilocalories, starchy staples and animal source foods (milk, egg, poultry, fish). However, in manuscript, cereals, pulses, fruit and vegetables and nuts have been involved in food system diagnosis but there is no data for whole grains in selected indicators and cutoffs for food systems diagnostics. As the part of healthy and sustainable diet whole grains and animal source foods can be involved to food system diagnosis.

- As mentioned in applying the diagnostics in Greece, Mediterranean diet is considered as a healthy and sustainable diet model to improve human and planetary health. So, Mediterranean diet can be discussed more deeply in Greece case study.

Reviewer #2: Manuscript is interesting, quite novel and well written and presented. Conclusions are supported by results and references support methods used. Some minor revision is needed. Please check some queries below to be answered:

1. Authors say: L. 340 However, Greece’s national pulse supply is just 14 grams per person per day, indicating a likely

challenge area, while other Mediterranean countries, including Italy and Spain, are 14 and 15 grams. Since amount are the same for these countries are these countries challenge area too?

2. why authors give so much attention on pulses? they are cheap and easy to cultivate but most people do not like them since they cause bloating bloating in the abdomen and reduce vitamin and metals and trace metals absorption. Pulse flour is an alternative to be included in diet. Is there any data on this? In most EU countries including Greece there is not EU subsidy to farmers for pulses as it is for other crops, that is why farmers are not invest in pulses. Also there is a lot of cheap import from neighbour countries.

3. Fig 2 needs to be explained more, what are the colors stand for, what is density for?

4. Please increase all figures pixel quality, not visible enough.

minor spelling mistakes. Minor revision is needed.

Reviewer #3: Dear authors,

I congratulate you regarding their work.

Best Regards,

6. PLOS authors have the option to publish the peer review history of their article (what does this mean?). If published, this will include your full peer review and any attached files.

Reviewer #1: No

Reviewer #2: No

Reviewer #3: No

---

## [Author Response · Author response to Decision Letter 0]

22 May 2022

PONE-D-22-10677

Diagnosing the performance of food systems to increase accountability towards healthy diets and environmental sustainability 

Response to reviewers

Dear Dr. Fatih Oz, 

Thank you for giving us the opportunity to submit a revised draft of the manuscript “Diagnosing the performance of food systems to increase accountability towards healthy diets and environmental sustainability” for publication in PLOS ONE. We appreciate the time and effort you and the reviewers dedicated to providing feedback on our manuscript and are grateful for the insightful comments and valuable improvements to our paper. We have incorporated most of the suggestions made by the reviewers. Those changes are detailed in track changes within the manuscript. Please see the table below for a point-by-point response to the reviewers’ comments and concerns with added text in red. All page numbers refer to the revised manuscript file with tracked changes. 

Reviewer 1 

Comment Revision

As known, generally agreed healthy diet constitutes are adequate amounts of fruit and vegetables, whole grains, legumes and nuts besides sufficient but not excessive intake of kilocalories, starchy staples and animal source foods (milk, egg, poultry, fish). However, in manuscript, cereals, pulses, fruit and vegetables and nuts have been involved in food system diagnosis but there is no data for whole grains in selected indicators and cutoffs for food systems diagnostics. As the part of healthy and sustainable diet whole grains and animal source foods can be involved to food system diagnosis. We agree with this comment. On page 6 of the manuscript, we explain that the only dietary intake indicators currently available across countries are for infants age 6-23 months and have added the following clarification, lines 120-124: “Dietary measures for other age groups (school-aged children, adults, and adolescents) do not currently meet the geographic distribution requirements to be included in the diagnostic approach, but diet quality data currently being collected by the Gallup World Poll and DHS will be added as soon as they are available, covering indicators of dietary adequacy and NCD risk factors in the general population (38).” 

Within the data for the 6-23 mo age range, there are no data collected on whole grains; there is an indicator of animal source food consumption, which is included as a diagnostic indicator (indicator #22). For the general population, we note that diet data are an important gap (line 456). We have added a sentence to the discussion, lines 460-466: “These outcomes will include the minimum dietary diversity for women of reproductive age (MDD-W); an indicator of consumption of all five recommended food groups (vegetables; fruits; pulses, nuts, and seeds; animal source foods; and starchy staples); and indicators of risk factors for NCDs defined within WHO and other global recommendations, including consumption of adequate fruits and vegetables; whole grains; pulses, nuts, and seeds; and fiber and limited consumption of free sugar, salt, fat, saturated fat, and red and processed meat.”

As mentioned in applying the diagnostics in Greece, Mediterranean diet is considered as a healthy and sustainable diet model to improve human and planetary health. So, Mediterranean diet can be discussed more deeply in Greece case study. We appreciate this comment and the importance of the Mediterranean diet; we have added references to literature that shows the healthiness and sustainability of the Mediterranean diet on page 22, lines 351-355, references 47, 48, and 49:

Willett, W.C., 2006. The Mediterranean diet: science and practice. Public health nutrition, 9(1a), pp.105-110.

Martínez-González, M.A., Salas-Salvadó, J., Estruch, R., Corella, D., Fitó, M., Ros, E. and Predimed Investigators, 2015. Benefits of the Mediterranean diet: insights from the PREDIMED study. Progress in cardiovascular diseases, 58(1), pp.50-60.

WHO, 2018. Healthy Diet Fact Sheet.

Reviewer 2 

Authors say: L. 340 However, Greece’s national pulse supply is just 14 grams per person per day, indicating a likely

challenge area, while other Mediterranean countries, including Italy and Spain, are 14 and 15 grams. Since amount are the same for these countries are these countries challenge area too? Thank you for this comment. That is correct, pulse supply is a likely challenge area for many countries including the ones listed. We have added language to make this more clear, lines 355-360: “However, Greece’s national pulse supply is just 14 grams per person per day, indicating a likely challenge area, while other Mediterranean countries, including Italy and Spain, are 14 and 15 grams, respectively, and France is just 4.7 grams per person per day, indicating it is a likely challenge areas for all of these countries.”

Why authors give so much attention on pulses? they are cheap and easy to cultivate but most people do not like them since they cause bloating in the abdomen and reduce vitamin and metals and trace metals absorption. Pulse flour is an alternative to be included in diet. Is there any data on this? In most EU countries including Greece there is not EU subsidy to farmers for pulses as it is for other crops, that is why farmers are not invest in pulses. Also there is a lot of cheap import from neighbour countries. We appreciate this comment. We have added additional data on fruit and vegetable supply in Greece that indicates it is not a likely challenge area, lines 353-355: “Greece has 747 grams of fruits and vegetables available per person per day, an abundant supply making it likely that most people in Greece would be able to access at least 400 grams of fruits and vegetables per day, the WHO-defined minimum.”

We have also added additional data on dietary patterns in Greece to further elaborate why we have pinpointed pulses as a relevant food systems challenge for both health and environmental outcomes, lines 363-371: “DQQ data from the Global Diet Quality Project indicate that in Greece, pulses are indeed a dietary gap, with just 18% of a nationally representative sample having consumed pulses in the day prior to the survey; this is coupled with relatively high consumption of red meat (44%) and processed meat (23%), and in contrast to high consumption of fruits and vegetables (95%) (38). These diet data indicate that higher pulse consumption could substitute for some red and processed meat consumption, with co-benefits for NCD risk and environmental impact. In addition to the low physical supply, low pulse consumption could be brought on by unaffordability of pulses; however, in Greece the cost of pulses relative to starchy foods is cheap, indicating that cost is less likely to be a contributor.”

Fig 2 needs to be explained more, what are the colors stand for, what is density for? Thank you for this comment. We have added a legend to Figure 2 to explain the figure further and what the colors and density stand for, lines 161-167: “Fig 2. Examples of diagnose thresholds for A) unimodal and B) bimodal indicator frequency distributions. Density plots of the data distribution for A) vegetable losses (% of domestic supply) and B) prevalence of adult obesity with colors indicating cutoffs for diagnostic thresholds: green indicates an unlikely challenge area, yellow indicates a potential challenge area, and red indicates a likely challenge area. Density plots are similar to histograms but use kernel density smoothing, rather than bins, to present a continuous distribution of the data. The peaks of each plot represent where the highest number of observations exists in the data.” 

We have also added the follow text to clarify as well, lines 150-154: “Where no cutoffs have been published and no normative values exist, the relative values of country data points can be compared as relatively higher or lower. For each indicator, density plots, a variation of histograms, were used to examine the distribution of data, using the data assembled on the FSD (S3 Table). A density plot was chosen over a histogram to view a smoothed distribution of the data using kernel density estimation.”

Please increase all figures pixel quality, not visible enough. We appreciate this comment. We have included higher resolution figures with our revisions.

Minor spelling mistakes Thank you for this comment. We have done another copy edit of the resubmitted manuscript and hope to have caught any remaining spelling and grammar mistakes.

Reviewer 3 

I congratulate you regarding their work. We appreciate your supportive comment.

We look forward to hearing from you in due time regarding our submission and to respond to any further questions and comments you may have.

Sincerely,

Rebecca McLaren 

May 22, 2022

---

## [Editor Report · Decision Letter 1]

16 Jun 2022

Diagnosing the performance of food systems to increase accountability towards healthy diets and environmental sustainability

PONE-D-22-10677R1

Dear Dr. Fanzo,

We’re pleased to inform you that your manuscript has been judged scientifically suitable for publication and will be formally accepted for publication once it meets all outstanding technical requirements.

Kind regards,

Fatih Oz, Ph.D.

Academic Editor

PLOS ONE

Additional Editor Comments (optional):

Dear Fanzo,

I am pleased to confirm that your paper "Diagnosing the performance of food systems to increase accountability towards healthy diets and environmental sustainability" has been accepted for publication in Plos One.
---

## [Editor Report · Acceptance letter]

20 Jul 2022

PONE-D-22-10677R1 

Diagnosing the performance of food systems to increase accountability toward healthy diets and environmental sustainability 

Dear Dr. Fanzo:

I'm pleased to inform you that your manuscript has been deemed suitable for publication in PLOS ONE. Congratulations! Your manuscript is now with our production department. 

Kind regards, 

on behalf of

Professor Fatih Oz 

Academic Editor

PLOS ONE